# Effect of Ginger Root Extract on Intestinal Oxidative Status and Mucosal Morphometrics in Broiler Chickens

**DOI:** 10.3390/ani14071084

**Published:** 2024-04-03

**Authors:** Oluwabunmi O. Apalowo, Radiah C. Minor, Adedeji O. Adetunji, Deji A. Ekunseitan, Yewande O. Fasina

**Affiliations:** 1Department of Animal Science, North Carolina Agricultural and Technical State University, Greensboro, NC 27411, USA; ooapalowo@aggies.ncat.edu (O.O.A.); rcminor@ncat.edu (R.C.M.);; 2Department of Agriculture, University of Arkansas at Pine Bluff, Pine Bluff, AR 71601, USA

**Keywords:** ginger root extract, intestinal oxidative status, mucosal morphometrics, biological antioxidant potential, reactive oxygen metabolites

## Abstract

**Simple Summary:**

Due to inadequate feed use, poor performance, low feed conversion ratios, changeable retail prices, and numerous illnesses, there is a rise in the use of antibacterial growth promoters. Antimicrobial resistance is a major issue in the industry; hence, novel approaches are needed. Ginger root extract replaces antibiotics as a long-term chicken productivity treatment. This study investigated ginger root extract’s influence on intestine oxidative status and mucosal growth in broiler chickens. At 0.75%, ginger root extract boosts chicken growth, while at 1.5%, it lowers gut oxidative stress and promotes mucosa growth.

**Abstract:**

This study was designed to assess the effect of ginger root extract (GRE) supplementation on the oxidative status and intestinal mucosal development in broiler chickens for 6 weeks. Day-old chicks (Ross 708 strain, *n* = 432) were distributed into six treatments with six replicate of twelve birds each: Negative CON (basal), MX (basal diet + bacitracin methylene disalicylate (BMD) 0.055 g/kg diet), GRE-1 (basal diet + 0.375% GRE), GRE-2 (basal diet + 0.75% GRE), GRE-3 (basal diet + 1.5% GRE), GRE-4 (basal diet + 3% GRE). Growth indices, goblets cell count, mucin (MUC2) in ileum tissue, antioxidant (SOD, CAT, and GPX) in ileum and liver, biological antioxidant potential (BAP), and reactive oxygen metabolite level in blood and intestinal villi measurement were determined. Body weight (BW) was highest (*p* < 0.05) in all groups except GRE-4, body weight gain (BWG) was best in GRE-1, while FCR was least in all groups except GRE-4. Optimum MUC2 gene expression, SOD, CAT, blood antioxidants, and intestinal morphometric values were observed in GRE-3. The inclusion of ginger root extract up to 1.5% improved growth and reduced oxidative stress while enhancing mucosal development in broiler chicks.

## 1. Introduction

Large-scale, intensive poultry farming exposes birds to numerous stressful conditions and diseases coupled with adverse changes in climate, resulting in devastating financial consequences [1]. Antibiotics are used as growth promoters in animal diets and have been the go-to in most developed and developing countries. The abuse of antibiotics resulted in the emergence and spread of antimicrobial resistance and, thereby, is classified as a global health concern. There are continuous calls for its restriction due to the emergence of antibiotic-resistant human pathogens [2] in the food chain, necessitating the need for viable plant-based alternatives. Natural alternatives such as ginger, garlic prebiotics, organic acids, plant extracts, etheric oils, and immune stimulants have been employed in recent years to increase performance, maintain chicken output, prevent and control enteric infections, and reduce antibiotic use in poultry production [2].

Ginger (*Zingiber* officinale Roscoe, *Zingiberaceae*) is an important crop grown mostly in Central Asia, China, India, and Pakistan and sold all over the world [3]. Ginger is widely used as a spice and in traditional medicine to treat a wide range of ailments [4]. Antioxidant, antimicrobial, and pharmacological capabilities are among the many potentials exhibited by bioactive compounds present in ginger root. Ginger is beneficial as a dietary supplement for broiler chickens due to its powerful antioxidant activity, which boosts the immunological response [5]. Ginger contains compounds such as shogaol, ginger-diol, gingerdione, and gingerols [6,7]. These compounds have been studied for their potential effects on the digestive system. In particular, gingerols, as a primary bioactive compound of ginger, have been associated with anti-inflammatory and antioxidant properties, which may contribute to gastrointestinal health, where the small intestine plays a crucial role in the digestion and absorption of nutrients in poultry [7]. Alterations to this organ may influence the function of other organs and systems in the organism [8] since feed ingredients are digested into simple molecules such as free short peptides, amino acids, free fatty acids, and monosaccharides [9]. These molecules are absorbed in the duodenum, jejunum, and ileum and then transferred to other tissues by blood circulation [10]. The gastrointestinal tract contains a mucous membrane that is protected by mucus, which is a thick, gel-like substance that covers the mucous membrane called mucus [11]. Goblet cells produce secretory mucin 2 (MUC2), a significant component of intestinal mucus that comes into direct contact with gut microorganisms [12]. The mucus-gel layer covers the intestinal mucosa and serves as the first line of defense against bacteria [13] and also as a protection and maintenance mechanism of the intestinal tract. The potential of ginger have been documented in poultry studies to explore its impact on the health and performance of poultry.

Oxidative status/stress in broiler chickens is influenced by environmental, dietary, microbiological, and management factors and is either beneficial or detrimental to poultry health and production [14]. In cells and tissues, oxidative stress results in lipid peroxidation, protein nitration, DNA damage, and death when free radical generation and natural antioxidant defenses are out of balance. In response to increased redox stress, oxidative stress indicators in poultry are molecules of the antioxidant system that change (reduce) [15]. Enzymes such as superoxide dismutase (SOD), catalase (CAT), and glutathione peroxidase (GPX) are plentiful in the small intestine, particularly throughout its development [16,17]. These enzymes serve as the body’s main defense mechanism against oxidative stress. Preserving gut health is an essential function of these antioxidants. Nonetheless, intestinal maturation may be impeded, and the concentrations of these endogenous defenses may be diminished if feed intake is reduced as a result of nutrient malnutrition [18]. The decline in antioxidant levels not only undermines the resistance of the small intestine to oxidative stress but also magnifies its effects on the liver, which is a vital organ in metabolic operations [19]. Hence, the health of chickens as a whole is impacted by variations in antioxidant levels that arise from decreased feed consumption, as these oscillations impact hepatic and intestinal functions.

However, we hypothesize that ginger root extract can reduce intestinal oxidative stress and enhance intestinal mucosal development in broiler chicks. The purpose of this research was to determine the optimum dietary ginger root extract concentration that would reduce oxidative stress while enhancing mucosal development in broiler chicks. Growth performance indicators were determined, and intestinal villi morphometrics (villi measurement), goblet cell density, mucin (MUC2) and antioxidant genes (SOD, CAT, and GPX), biological antioxidant potential, and reactive oxygen metabolites were measured on day 6, 13, 27, and 41 to determine the oxidative status and mucosal development in broiler chicks, respectively.

## 2. Materials and Methods

All the procedures used in the experiment were approved by NCATSU’s Institutional Animal Care and Use Committee (IACUC, Protocol # 20-003).

### 2.1. Experimental Design, Animals, Housing, and Experimental Diets

A total of 432 day-old male broiler chicks (Ross 708 strain) were completely randomized to six dietary treatment groups. Each treatment contained six replicate pens, each with twelve chicks allocated to each pen. The trial lasted for 42 days (6 weeks). Treatment 1 (CON) consisted of chicks fed unmedicated corn–soybean meal (SBM). Treatment 2 (MX) consisted of chicks given unmedicated corn–SBM basal with bacitracin methylene disalicylate (BMD) added at 0.055 g/kg diet. Treatment 3 (GRE-1), treatment 4 (GRE-2), treatment 5 (GRE-3), and treatment 6 (GRE-4) consisted of chicks given unmedicated corn–SBM basal with dietary ginger root extract added at 0.375%, 0.75%, 1.5%, and 3% levels of the diet, respectively. Ginger root extract was purchased commercially from Sabinsa Corporation, East Windsor, NJ. The chicks spent their first 21 days in a battery cage and were later moved to a floor enclosure with a nipple drinker line, a hanging feeder, and clean, new litter. Diets (Table 1 and Table 2) used in experiments were designed so as to adhere to, meet, or slightly exceed nutrient requirements recommended in the Aviagen 2022 Ross broiler nutrition specification handbook. Starter diets were supplied as crumble from day 1 to day 21, and grower diets were fed as pellets from day 22 to day 42 of the experiment. During the 42-day trial, the birds had free access to food and water.

### 2.2. Growth Performance

The body weight (BW) and feed intake (FI) of the birds were measured, and these values were used to calculate body weight gain (BWG) and feed conversion ratio (FCR). Daily mortalities were observed and recorded.

### 2.3. Sample Collection

On days 6, 13, 27, and 41, one bird from each cage (*n* = 6 per treatment) was randomly sampled to collect small intestines before being preserved in 10% neutral-buffered formalin (Fisher Scientific, Pittsburgh, PA, USA) until tissue processing for downstream villi morphometric examination; each tissue piece was gently washed in cold saline (0.9% NaCl) to remove gut contents. The expression levels of antioxidant genes (superoxide dismutase (SOD), catalase (CAT), and glutathione peroxidase (GPX)) were determined by real-time quantitative reverse-transcription polymerase chain reaction, samples were collected for qRt-PCR, placed in a DNase- and RNase-free cryovial, snap frozen in liquid nitrogen, and stored at 80 °C until analysis (qRT-PCR). Each sample’s plasma was extracted in the Poultry Ethnomedicine Research Laboratory in Webb Hall at North Carolina Agricultural and Technical State University by centrifuging the blood for 10 min at 4200 rpm in a centrifuge (Thermo Fischer Scientific Inc., Waltham, MA, USA). The obtained plasma was frozen at −80 degrees Celsius for biological antioxidant potential (BAP) and Reactive Oxygen Metabolites analysis.

### 2.4. Intestinal Villi Morphometrics

Tissue samples of the duodenum, jejunum, and ileum of the sampled chicks were trimmed into transverse, longitudinal, and vertical pieces and placed in formalin. It is standard practice to embed the clipped tissues in paraffin and produce slides. For the 6 treatments (*n* = 6), 10 properly aligned villus-crypt units from each bird were measured morphometrically using the NIS Elements BR program (Nikon Instruments Inc., Melville, NY, USA). The following gastrointestinal parameters were measured: duodenum villi width (DVW), duodenum villi height (DVH), duodenum muscle layer (DML), duodenum crypt depth (DCD), and duodenum villi/crypt (DVC); jejunum villi width (JVW), jejunum villi height (JVH), jejunum muscle layer (JML), jejunum crypt depth (JCD) and jejunum villi/crypt (JVC) and ileum villi width (IVW), ileum villi height (IVH), ileum muscle layer (IML), ileum crypt depth (ICD) and ileum villi/crypt (IVC). The villus/crypt ratio was determined by dividing the villus height by the crypt depth for all villi and crypts that were measured [20]. The density of goblet cells (GCs) in ileal tissue was determined by counting GCs at 20× magnification, one per villus area. These values were calculated as an average across all intestinal segments for each bird.

### 2.5. Isolation and Quantification of RNA from Chicken Intestine

Tissue samples of about 30 to 40 mg were weighed by cutting a piece of each tissue into a weigh boat using forceps and scissors. Then, 700 µL of lysis solution from (Bio-Rad, Hercules, CA, USA) was pipetted into labeled 2 mL sterile Sarstedt tubes, and the weighed ileum tissue samples were placed on ice. The samples were homogenized using a bead-beater at 4800 rpm for 90 s. Tubes containing the homogenate were transferred into a centrifuge at 4 °C at 9000 rpm for 5 min. The supernatant was transferred into a new 2 mL capped microcentrifuge tube, and 700 µL of 60% ethanol was added to the supernatant and mixed thoroughly until no visible bilayer was observed. The RNA binding column was inserted into a 2 mL capless wash tube. The homogenized lysate was decanted into the RNA binding column and centrifuged at 9000 rpm for 1 min at 4 °C. The RNA binding column was removed from the wash tube, the filtrate was discarded from the wash tube, and the column was replaced in the same tube. After centrifuging the RNA binding column at 9000 rpm for 30 s, the wash solution was discarded from the wash tube, and the column was replaced in the same wash tube. A total of 80 µL of diluted DNase was added to the membrane stack at the bottom of each column, and the digest was allowed to incubate at room temperature for 25 min. The 700 µL high stringency wash solution was added to the RNA binding column and centrifuged at 9000 rpm for 1 min before discarding the high stringency solution and replacing the column with the same tube of low stringency wash solution was added to the RNA binding column and centrifuged at 9000 rpm for 2 min. The low-stringency solution was discarded and replaced with the column in the same tube. It was centrifuged for another 2 min at a speed of 9000 rpm to remove the residual wash solution. The RNA binding column was transferred into a 1.5 mL capped microcentrifuge tube, and 80 µL of the elution solution was added onto the membrane stack at the bottom of the RNA binding column. The solution was allowed to saturate the membrane for 1 min, then centrifuged at 9000 rpm for 2 min to elute the total RNA. The optical density of each sample was 260/280 nm, and 2 µL of blank was added to two locations for proprietary Bio-Cell measurements using the BioTek Synergy H1 microplate reader (BioRad Laboratories Inc., Hercules, CA, USA).

### 2.6. cDNA Synthesis & Quantitative Real-Time PCR Analysis

The expression level of genes for MUC2 and antioxidant enzymes (i.e., SOD, CAT, GPX) in the ileal tissue was determined using the isolated RNA samples obtained from the section.

First, cDNA was synthesized from the RNA samples using iScript Reverse Transcription Supermix (BioRad, USA). A total of 4 µL of 5 × iScript reaction mix (RT/No-RT) was added to each calculated sample. Primers used in the qRT-PCR analysis were designed in our laboratory and are presented in Table 3. The qRT-PCR was performed using SYBR green supermix. Bio-Rad CFX connects with a working sample of the 3 µL of each cDNA sample, 10 µL of SSO Advanced Universal SYBR Supermix (Bio-Rad), 1 µL (0.5 µL forward and reverse) of each primer, 3 µL of each cDNA template, 6 µL of free DNAse/RNAse water, totaling 20 µL which was distributed in triplicate to a 96-well plate and sealed using BioRad Microseal ‘B’ seal (an optically clear heat seal), centrifuged and placed inside a CFX connect machine running at a temperature of 60 °C. ΔΔCT was used in calculating the fold changes for the upregulation or downregulation of genes.

### 2.7. Biological Antioxidant Potential and Reactive Oxygen Metabolites

A d-ROMs and BAP test kit were used to perform biological antioxidant potential and reactive oxygen metabolites tests on free DUO (DIACRON Research and Diagnostics, Grosseto, Italy). To conduct the reactive oxygen metabolites assay, 20 µL of plasma was pipetted into a thermostated cuvette, and then 20 µL of d-ROM reagent was pipetted into the same cuvette using separate tips. A little stir was given, then the cuvette was placed in the reading cell, where it was automatically started and incubated for 3 min and 2 min for kinetic reading.

To perform a biological antioxidant potential test, 50 µL of the BAP reagent was pipetted into the thermostated cuvette, mixed gently, and placed in the reading cell, which automatically read for 5 s; then, 10µL of plasma was pipetted into the cuvette with the reagent, mixed gently, and placed in the reading cell for another 300 s.

### 2.8. Statistical Analysis

All experimental data were analyzed using one-way analysis of variance (SAS Institute, Cary, NC, USA, 2004), and mean differences were determined using the Duncan multiple range test. Statistical significance was assumed at the *p* = 0.05. The body weight, feed intake, feed conversion ratio, villus height, villus width, crypt depth, mucosal layer, goblet cell count, mucin-2 (MUC2) content, and catalase (CAT) and glutathione peroxidase (GPX), biological antioxidant potential, and reactive oxygen metabolites of the broiler diet are all dependent variables in this study. Additionally, linear regressions were performed to model and analyze the relationships between the variables assessed and GRE supplementation. The MX treatment was not considered in the regression analysis.

## 3. Results

### 3.1. Growth Performance

Table 4 shows the cumulative growth performance data of broiler chickens from d1 to d42 supplemented with dietary GRE at 0.375%, 0.75%, 1.5%, and 3% (GRE-1, GRE-2, GRE-3, and GRE-4, respectively). GRE-4 (BW = 2.923) was the least compared to CON, MX, and other treatment groups. FCR was higher and poorer in GRE-4 compared to other treatment groups. No mortality was reported in the study. There was no significant difference in FI amongst all treatment groups. 

### 3.2. Intestinal Villi Morphometrics and Goblet Cell Density

Analysis of broiler chicken intestinal morphometrics: duodenum, jejunum, and ileum revealed a difference in CON, MX, and other GRE levels on days 6 and 13.

#### 3.2.1. Effect of Dietary GRE Supplementation on Intestinal Villi

Among the characteristics measured on days 6 and 13 (Table 5), only duodenal villus weight (DVW) was significantly different (*p* < 0.05). On day 6, the DVW for the GRE-supplemented treatment at 1.5% (GRE-3: 136.96) was greater than the DVW for the MX-supplemented treatment (80.56); i.e., GRE-3 is the highest amongst the treatment groups.

In the jejunum (Table 6), only JML on day 6 and the JVC ratio on day 13 were affected by GRE supplementation. There was no apparent trend in the differences observed in JML values among treatments on d 6; however, the lowest values were observed in GRE-2 and GRE-4. At day 13, GRE-3 had a much higher JVC ratio (6.16), and GRE-4 had a slightly higher ratio (5.03), both of which were higher than the ratio seen for MX (3.89) and other treatments. Significant differences (*p* < 0.05) were found in the JVC ratio for GRE-3.

All GRE-supplemented treatments in the ileum (Table 7) revealed increased IVW (GRE-1 = 144.24, GRE-2 = 127.23, GRE-3 = 116.92, and GRE-4 = 116.54) in comparison to CON (92.39) and MX (109). IML with GRE-2 (118.66) has the highest value among other treatment groups.

#### 3.2.2. Effect of GRE on Goblet Cell Density of Ileum Day 6 and 13

Table 8 presents a significant difference in goblet cell density between GRE and other treatments. However, GRE-3 and CON were the least among all GRE-supplemented diets and MX on day 6 (*p* < 0.05) in the ileum. On the other hand, on day 13, GRE-1 and GRE-2 exhibited the least value in all GRE treatments but were similar to CON and MX (*p* < 0.05).

### 3.3. Effect of GRE Supplementation on Selected Antioxidant Genes 

Figure 1 shows a substantial change in the expression of the MUC2 gene on GRE-2 and GRE-3 on day 6. On day 13, GRE-3 improved and upregulated the expression of MUC2 in the ileum sample compared to MX and other levels of GRE inclusion. 

Figure 2 shows that there were no significant differences in the ileum on d6 and d13, but the result shows that the activity of SOD was highly expressed on GRE-3 on d6 compared to d13, which states that on ileum of early birds’ activity of SOD is highly expressed. In the liver at day 27, GRE-1 and GRE-2 highly enhanced (*p* < 0.05) SOD activity compared to other treatment groups.

Figure 3 indicates that in ileum d6, GRE-3 upregulated the activity of CAT in a statistically similar manner to CON. On d13, the activity of CAT was highly expressed in GRE-supplemented groups (GRE- and GRE-4) and MX. On liver d27, among all GRE treatments, GRE-4 showed higher expression of CAT compared to CON and MX. 

On d6, MX and all GRE treatment groups were different from CON in ileum GPx expression, as depicted in Figure 4, and on d13, there was no significant difference among all treatments. However, on liver d27, only GRE-4 showed that GPx activity was highly upregulated compared to CON, MX, and other GRE groups.

On day 13, GR- 2 and GRE-3 had the highest biological antioxidant potential expression, comparable to CON and MX, but with lower antioxidant values in GRE-1 and GRE-4, while the least antioxidant values were observed in MX and GRE-4 on day 27 (Figure 5). On day 41, as the level of GRE supplementation increased, there was a reduction in the biological antioxidant potential, with the lowest value in GRE-4 and similar and highest values in CON and MX.

The reactive oxygen metabolite in plasma presented in Figure 6 revealed no difference (*p* > 0.05) at d13, d27, and d41, but there was a continual reduction trend in the Reactive Oxygen Metabolites values in GRE-3 and GRE-4.

Table 9 presents a linear regression analysis. The increasing supplementation of GRE linearly reduced FCR, BWG, and goblet cell density. The *p*-values of Y^1^ and Y^2^ (*p* <0.05) indicated a significant positive relationship between Y (dependent variable: BWG and FCR) and X (GRE supplementation). The regression of accessed indicators (goblet cell density, BAP, JVC, MUC2, and FCR) in the study against BWG revealed positive coefficients (Gb and JVC), suggesting a positive relationship, while negative coefficients (BAP, MUC2, and FCR) indicated a negative relationship with BWG.

## 4. Discussion

Ginger root extract has some bioactive components that influence growth performance and immune response due to their antioxidant, antimicrobial, and anti-inflammatory properties by neutralizing free radicals and activating antioxidant enzymes, mitigate inflammation by inhibiting pro-inflammatory pathways and enzymes, promote digestion through enzyme stimulation and gastroprotective effects, and modulate immune responses, fostering overall health [21]. Because GRE contains compounds that perform biological functions, we supplemented GRE in the broiler diet in a bid to investigate the effects of GRE on growth performance, its role in oxidative stress, and mucosal development. In this present study, GRE supplementation up to 0.75% improved body weight, body weight gain, and feed conversion ratio of broiler chickens in comparison to other GRE treatments. This was affirmed by the regression analysis carried out in the study, which revealed a linear reduction in BWG and FCR. Similar findings [9] indicate that broilers fed 2% dried ginger meal as a supplement consumed less feed than those fed the control diet. Improvement in birds’ body weight with the inclusion of ginger root extract could be attributed to its favorable effects on endogenous enzyme activity, intestinal microflora, and/or gut function [22], and it also indicates that ginger stimulates appetite and has digestive properties, enhancing feed intake and digestion with gingerol and shogaol as their active compounds [23]. This improved performance can be attributed to the presence of a proteolytic enzyme protease, zingibain, present in ginger [24] and reported to aid in the digestion of proteins through increased production of digestive enzymes in the body. Ginger has also been mentioned to aid in stimulating and increasing the body’s production of amylases and lipases needed for digestion via bioactive compounds present in it leading to nutrient digestibility and absorption to support growth in birds [25].

Morphometric evaluation is a useful tool for evaluating intestinal health since villi height positively correlates with intestinal absorption area, and cryptal depths correlate with intestinal renewal rate; an increase in villi height provides a larger surface area for absorption, while deeper crypts support the continuous renewal of absorptive cells [26]. Except for DVW, the morphometric parameters DVW, DVH, DML, DCD, and DVC in the duodenum did not alter significantly between day 6 and day 13. This suggests that birds fed GRE-3 (GRE 1.5% of the diet) in the early stages of development had a higher rate of digestion than birds fed diets supplemented with CON and MX. There was a positive correlation between the length and width of villi and the surface area available for nutrient absorption. This, in turn, enhances the overall performance and development of the birds [27]. Villus height in poultry is positively related to the intestine’s digestive and absorptive function and is an indicator of healthier birds with better gut health. Zhang et al. [7] reported that the optimal values of the intestine were attained between day 8 and day 17, as there was a dose-dependent increase in the concentration of 10-gingerdiols and 10-shogaol at 1.5% dose from week 2 to week 4. This could account for the positive response and increase observed in the GRE-3 supplementation, as birds were able to attain maximal allometric growth of the small intestine. Consequently, the obtained result indicated that the nutrients were effectively retained until day 13. The morphometric characteristics assessed for the jejunum indicated that birds fed GRE-3-supplemented diets had a greater JVC. This would suggest that the jejunum’s maturity caused the absorption rate to increase at this level of supplementation [28]. In addition, the ileum villi measurement on day 6 revealed that all GRE supplementation improved the width, while GRE-2 had a positive effect on the muscle layer of the villi. Therefore, it may have a function in enhancing the surface area and promoting the transport of nutrients, antigens, and fluids. Research has generally indicated that GRE contains active ingredients, including gingerol and shogaol, that support intestinal growth and validate ginger’s antibacterial and immunomodulatory qualities [23].

Goblet cells secrete mucin 2 (Muc2), a primary component of the mucus that coats the digestive tract and forms a protective barrier between pathogens and the intestinal epithelial cells; hence, they are crucial for chick health because the density and activity of intestinal goblet cells increases in the first week of age [29]. Mucin secretion is typically linked with enhanced mucin production, whereas prolonged mucin secretion may result in goblet cell depletion and reduced mucin synthesis [30]. The statistical increase in value of all treatments except for the CON group on day 6 represents an increase in the absorptive surface area of the intestine and, thus, an increased absorptive capacity in birds [31], while on day 13, higher goblet cell density was observed in GRE-3 and GRE-4. The positive increase reported in this study as a result of GRE supplementation enhanced the number of goblet cells and projected a defensive barrier of the intestinal mucus layer [30]. Moreover, gene expression data was upregulated in GRE-2 and GRE-3 on day 6, while on day 13, GRE-3 increased MUC2 expression compared to other treatments, which indicates that in the presence of stressors, goblet cells were able to release mucin to serve as a protective barrier against microbes and isolate the intestinal epithelium from the luminal cavity, defending against pathogens, maintaining gut health, and facilitating efficient absorption of nutrients [32].

Oxidative stress in cells and tissues originates from an imbalance between free radical generation and antioxidant defense [33], which leads to lipid peroxidation, protein nitration, DNA damage, and apoptosis. The oxidative state of broiler chickens was determined by evaluating antioxidant (SOD, CAT, and GPX) indices in the ileum (days 6 and 13) and liver (day 27). On days 6 and 13, the expression levels of SOD in the ileum were not substantially different between the treatment and control groups. SOD, being an essential vitagene, plays a pivotal role in cellular and bodily adaptation to many significant stressors [34] and in combating reactive oxygen species induced by diseases within the body [24,35]. However, on day 27, there was an increase in SOD expression in birds on GRE-3 supplementation. Increased SOD expression in GRE-3 in the current study is a preventive mechanism against oxidative stress [36], affirming the antioxidant capacity of ginger in reducing reactive oxygen species (ROS) through promoting SOD expression [34].

On day 13, ileum CAT expression in GRE treatment and MX increased considerably relative to the CON group. In the liver, d27 showed similarities in all treatments except for GRE-4, which could indicate that as birds age, the expression of CAT increases and reduces their susceptibility to lipid peroxidation [37]. During the latter period of growth, which is characterized by cell proliferation and muscle development, hydrogen peroxide buildup is enhanced [38]. Consequently, CAT seeks to reduce hydrogen peroxide buildup [39]. The addition of ginger to the meal revealed significant differences in the ileum at d6 and the liver at d27; GPX levels were upregulated, indicating that there was no oxidative stress in the system. This negates the report of [40], who observed that ginger-fed birds exhibited no significant reduction in GPX gene expression. The inclusion of GRE up to 1.5% will increase the abundance of SOD, CAT, and GPX, which are the first line of antioxidant defense to reduce oxidative damage in poultry. Ginger is widely regarded as the storehouse of antioxidants and contains zingerone, a methoxyphenol family shown to have a direct adaptogenic impact by shielding the intestinal smooth muscles from oxidative stress [41]. The abundance of GPX, CAT, and SOD illustrates its remarkable efficiency in scavenging reactive oxygen species (ROS), free radicals, peroxides, and other harmful oxidants. 

Hydroperoxides are a type of chemical oxidant species that are part of the larger category of reactive oxygen metabolites, and their blood content is measured by the d-ROMs test, which is regarded as the gold standard for total oxidative stress. The oxidation of several compounds, including glucosides, lipids, amino acids, peptides, proteins, and nucleotides, results in the formation of hydroperoxides [42]. Reactive oxygen metabolites were similar in all treatment groups; however, there was a decreasing trend observed in the GRE-3 and GRE-4 groups, with the values observed on day 41 in line with the normal range zone (20–24) reported by [43]. This implies that the inclusion of GRE up to GRE-3 helped maintain homeostasis within the system and reduced oxidative stress in birds without having a negative impact on the birds [43].

The BAP test is a photometric measurement of the plasma’s biological antioxidant potential, defined as the sample’s ability to convert ferric (Fe^3+^) iron to ferrous (Fe^2+^) iron. The biological antioxidant potential was highly expressed in all groups at d13, d27, and d41, except Mx at d27. The values were higher than the optimum value (>2200 µmol/L) reported by [44], which correlated with the plasma antioxidant effectiveness. Past research also shows that the relatively high concentrations of antioxidants observed in the study may be significant to meet up with the rapid metabolism associated with growth and development in broiler chickens [44], affirming the antioxidant and antiapoptotic properties of the bioactive compounds present in ginger [3].

## 5. Conclusions

Ginger root extract can be used as an alternative to conventional growth promoters (BMD) in poultry production to increase growth performance and intestinal development and minimize or neutralize oxidative stress in broiler chickens. The inclusion of up to 1.5% GRE in the diet of broiler chickens improved feed efficiency, promoted intestinal activity, and reduced the risk of intestinal integrity disturbance and cell damage induced by oxidative stress.

## Figures and Tables

**Figure 1 animals-14-01084-f001:**
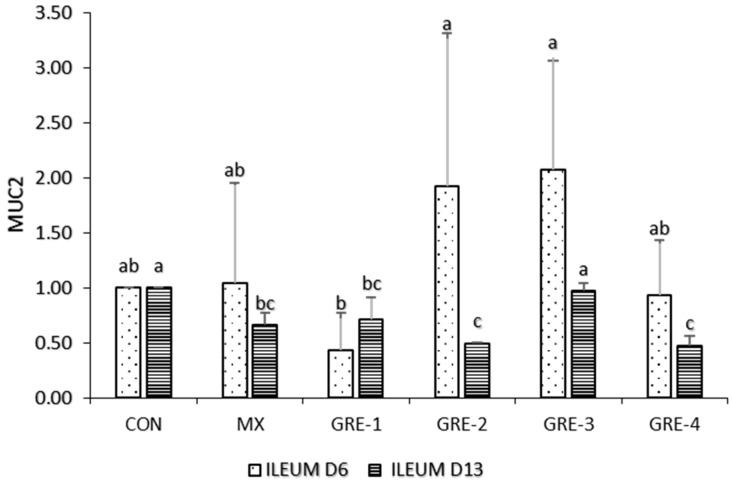
Effect of GRE supplementation on ileum tissue with MUC2. ^a–c^ Mean values bearing different superscript letters are significantly different (*p* < 0.05).

**Figure 2 animals-14-01084-f002:**
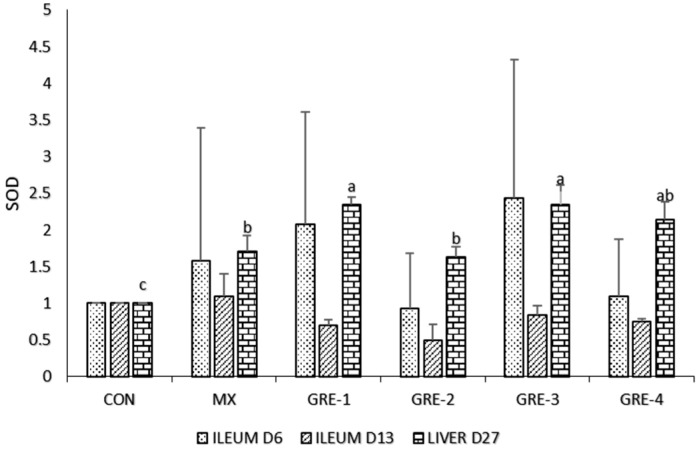
Effect of GRE supplementation on ileum tissue and liver with SOD. ^a–c^ Mean values bearing different superscript letters are significantly different (*p* < 0.05).

**Figure 3 animals-14-01084-f003:**
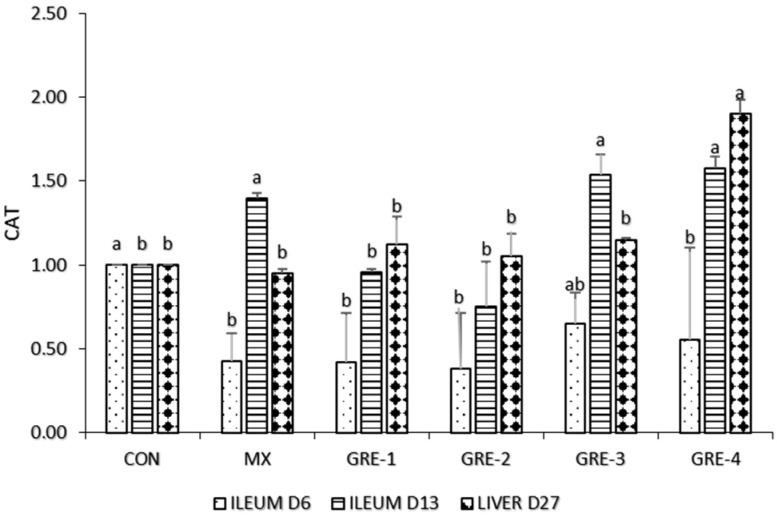
Effect of GRE supplementation on ileum tissue and liver with CAT. ^a,b^ Mean values bearing different superscript letters are significantly different (*p* < 0.05).

**Figure 4 animals-14-01084-f004:**
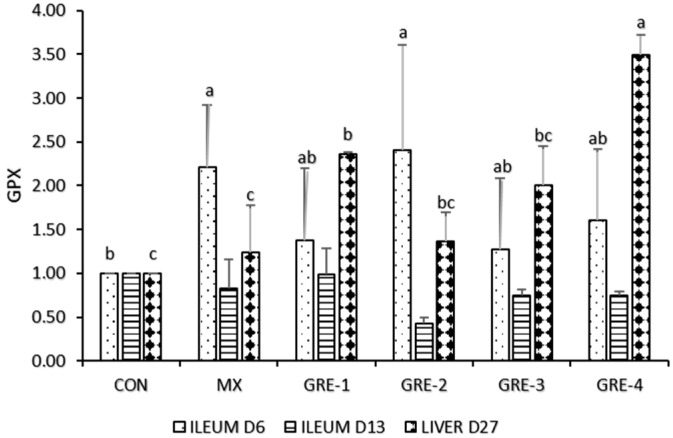
Effect of GRE supplementation on ileum tissue and liver with GPX. ^a–c^ Mean values bearing different superscript letters are significantly different (*p* < 0.05).

**Figure 5 animals-14-01084-f005:**
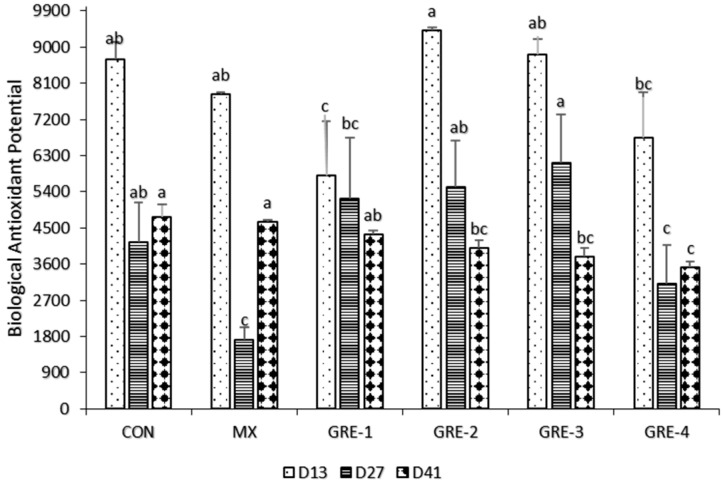
Effect of GRE supplementation on Biological Antioxidant Potential (BAP). ^a–c^ Mean values bearing different superscript letters are significantly different (*p* < 0.05).

**Figure 6 animals-14-01084-f006:**
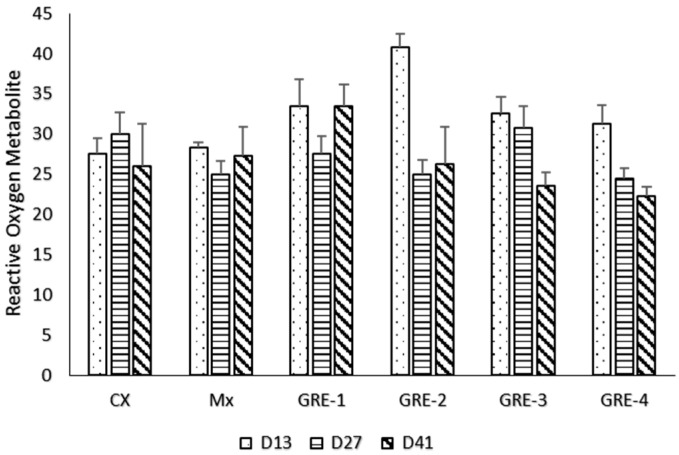
The effect of GRE supplement on Reactive Oxygen Metabolite.

**Table 1 animals-14-01084-t001:** Composition of Experimental Starter Diets (%).

Ingredients	Control	MX	GRE-1	GRE-2	GRE-3	GRE-4
Corn	51.46	51.45	55.72	56.34	55.90	59.40
Soybean meal	40.39	40.39	40.23	40.08	39.78	39.17
NCAT GRE ^1^	---	---	0.38	0.75	1.50	3.00
Poultry fat	3.64	3.64	3.63	3.61	3.59	3.53
Limestone	1.07	1.07	1.07	1.06	1.05	1.04
MDP ^2^	2.03	2.03	2.03	2.01	1.99	1.97
Salt NaCl	0.40	0.40	0.40	0.40	0.39	0.39
Sodium bicarbonate	0.02	0.02	0.02	0.02	0.02	0.02
L-Lysine HCl 98%	0.13	0.13	0.13	0.13	0.13	0.13
DL-Methionine 99.0%	0.34	0.34	0.34	0.33	0.33	0.33
L-Threonine 98.5%	0.11	0.11	0.11	0.11	0.11	0.11
NCSU Poultry Vitamin Premix ^3^	0.05	0.05	0.05	0.05	0.05	0.05
NCSU Poultry Mineral Premix ^4^	0.20	0.20	0.19	0.20	0.20	0.19
Bacitracin (Antibiotic, g/kg)	---	0.055	---	---	---	---
Choline chloride 60%	0.10	0.10	0.10	0.10	0.10	0.10
Selenium Premix	0.05	0.05	0.05	0.05	0.05	0.05
Analyzed nutrient composition ^5^						
Metabolizable energy (Kcal/kg)	3150	3155	3131	3197	3142	3201
Crude Protein, %	22.13	24.88	23.69	24.69	23.69	23.63
Crude Fat, %	5.15	5.47	5.71	5.30	6.00	6.87
Crude Fiber, %	2.5	2.7	2.3	2.4	2.6	2.4
Ash, %	5.59	5.46	5.21	5.47	5.67	5.47
Calculated nutrient composition						
Total Sulfur Amino Acids, %	1.03	1.03	1.03	0.94	0.95	0.92
Lysine, %	1.42	1.42	1.41	1.26	1.25	1.23
Calcium, %	0.96	0.96	0.96	0.74	0.74	0.73
Available phosphorus, %	0.48	0.48	0.48	0.44	0.43	0.43

^1^ NC A&T Ginger Root Extract (GRE), ^2^ Mono-Dicalcium phosphate, ^3^ Vitamin Premix, supplied per kilogram of diet: Vitamin A (6600 IU), Vitamin D (1980 IU), Vitamin E (33 IU), Vitamin B12 (0.02 mg), Biotin (0.13 mg), Menadione (1.98 mg), Thiamine (1.98 mg), Riboflavin (6.60 mg), d-Pantothenic Acid (11.0 mg), Vitamin B6 (3.96 mg), Niacin (55.0 mg), Folic Acid (1.1 mg). ^4^ Mineral Premix, supplied per kilogram of diet: Manganese (Mn), 60 mg; Zinc (Zn), 60 mg; Iron (Fe), 40 mg; Copper (Cu), 5 mg; Iodine (I), 1.2 mg; Cobalt (Co), 0.5 mg. ^5^ Experimental diets were analyzed for proximate nutrient composition by Eurofins Scientific Inc. Nutrient Analysis Center, 2200 Ritten-house Street, Suite 150, Des Moines, IA 50321, USA.

**Table 2 animals-14-01084-t002:** Composition of Experimental Grower Diets (%).

Ingredients	Control	MX	GRE-1	GRE-2	GRE-3	GRE-4
Corn	56.77	56.76	56.55	56.34	55.91	55.06
Soybean meal	35.43	35.43	35.29	35.16	34.89	34.37
NCAT GRE ^1^	---	---	0.38	0.75	1.50	3.00
Poultry fat	4.00	4.00	3.98	3.97	3.94	3.88
Limestone	0.64	0.64	0.64	0.64	0.63	0.62
MDP ^2^	1.84	1.84	1.83	1.83	1.81	1.80
Salt NaCl	0.40	0.40	0.40	0.40	0.39	0.39
Sodium bicarbonate	0.02	0.02	0.02	0.02	0.02	0.02
L-Lysine HCl 98%	0.11	0.11	0.11	0.11	0.11	0.11
DL-Methionine 99.0%	0.30	0.30	0.30	0.30	0.30	0.29
L-Threonine 98.5%	0.09	0.09	0.09	0.09	0.09	0.09
NCSU Poultry Vitamin Premix ^3^	0.05	0.05	0.05	0.05	0.05	0.05
NCSU Poultry Mineral Premix ^4^	0.20	0.20	0.20	0.20	0.20	0.20
Bacitracin (Antibiotic, g/kg)	---	0.055	---	---	---	---
Choline chloride 60%	0.10	0.10	0.10	0.10	0.10	0.10
Selenium Premix	0.05	0.05	0.05	0.05	0.05	0.05
Analyzed nutrient composition ^5^						
Metabolizable energy (Kcal/kg)	3109	3232	3131	3195	3142	3201
Crude Protein, %	22.13	22.94	23.69	22.00	21.94	20.88
Crude Fat, %	5.36	8.22	5.80	6.57	6.61	7.31
Crude Fiber, %	2.6	2.6	2.6	2.6	2.7	2.7
Ash, %	5.18	5.19	5.46	5.35	5.03	4.82
Calculated nutrient composition						
Total Sulfur Amino Acids, %	1.03	1.03	1.03	0.94	0.94	0.91
Lysine, %	1.42	1.42	1.40	1.26	1.25	1.23
Calcium, %	0.96	0.96	0.96	0.74	0.74	0.73
Available phosphorus, %	0.48	0.48	0.48	0.44	0.43	0.43

^1^ NC A&T Ginger Root Extract (GRE), ^2^ Mono-Dicalcium phosphate. ^3^ Vitamin Premix, supplied per kilogram of diet: Vitamin A (6600 IU), Vitamin D (1980 IU), Vitamin E (33 IU), Vitamin B12 (0.02 mg), Biotin (0.13 mg), Menadione (1.98 mg), Thiamine (1.98 mg), Riboflavin (6.60 mg), d-Pantothenic Acid (11.0 mg), Vitamin B6 (3.96 mg), Niacin (55.0 mg), Folic Acid (1.1 mg). ^4^ Mineral Premix, supplied per kilogram of diet: Manganese (Mn), 60 mg; Zinc (Zn), 60 mg; Iron (Fe), 40 mg; Copper (Cu), 5 mg; Iodine (I), 1.2 mg; Cobalt (Co), 0.5 mg. ^5^ Experimental diets were analyzed for proximate nutrient composition by Eurofins Scientific Inc. Nutrient Analysis Center, 2200 Ritten-house Street, Suite 150, Des Moines, IA 50321, USA.

**Table 3 animals-14-01084-t003:** Specific primer of selected genes and the housekeeping gene.

Gene	Forward Primer (5′-3′)	Reverse Sequence (5′-3′)
MUC2	ACGTGTGTGCCCATCTCCAA	GGGGACGCGTTGCAATCAAA
SOD	GGAGCGGGCCAGTAAAGGTTA	TCACATTGCCGAGGTCACCC
CAT	TGCAAAATGGCTGACGGACG	AGCACCAGTGGTCAAGGCAT
GPX	CTACAGCCGCCACTTCGAGA	AGAGGTGCGGGCTTTCCTTT
GAPDH	GAAGCTTACTGGAATGGCTTTCC	CGGCAGGTCAGGTCAACAA

**Table 4 animals-14-01084-t004:** Effect of dietary ginger root extract supplementation on growth performance of broiler chicks (d1 to 42).

Treatments 1	BW (kg/Bird)^3^	BWG (kg/Bird)	FI (kg/Bird)	FCR (kg/kg)
CON	3.266 ^a^	3.230 ^ab^	5.490	1.702 ^b^
MX	3.325 ^a^	3.310 ^ab^	5.500	1.662 ^b^
GRE-1	3.376 ^a^	3.375 ^a^	5.637	1.669 ^b^
GRE-2	3.273 ^a^	3.260 ^ab^	5.597	1.717 ^b^
GRE-3	3.143 ^ab^	3.108 ^bc^	5.327	1.713 ^b^
GRE-4	2.923 ^b^	2.901 ^c^	5.278	1.821 ^a^
SEM	0.057	0.072	0.101	0.020
*p*-value	<0.0001	<0.0001	0.1068	<0.0001

^a–c^ Mean values bearing different superscript letters within a column are significantly different (*p* < 0.05). CON (unmedicated basal diet), MX (basal diet with bacitracin methylene disalicylate), GRE-1 (basal diet with 0.375% ginger), GRE-2 (basal diet with 0.75% ginger), GRE-3 (basal diet with 1.5% ginger), GRE-4 (basal diet with 3% ginger). BW (body weight), BWG (body weight gain), FI (feed intake), FCR (feed conversion ratio).

**Table 5 animals-14-01084-t005:** Effect of ginger root extract supplementation on duodenal villi morphometrics in broiler chicks.

Treatments	DVW	DVH	DML	DCD	DVC
Day 6					
CON	92.11 ^bc^	635.84	97.92	100.57	6.46
MX	80.56 ^c^	591.70	112.60	99.02	6.16
GRE-1	132.59 ^ab^	684.40	103.92	120.36	5.82
GRE-2	115.46 ^abc^	798.38	113.28	110.40	7.53
GRE-3	136.96 ^a^	764.28	111.81	115.29	6.92
GRE-4	123.56 ^ab^	643.39	97.48	99.83	6.66
SEM	10.54	48.93	11.18	6.36	0.41
*p*-Value	0.0352	0.1676	0.9121	0.2850	0.2621
Day 13					
CON	107.74	920.10	144.71	118.29	118.29
MX	155.04	911.20	157.96	141.79	141.79
GRE-1	124.77	932.70	157.38	135.56	135.56
GRE-2	137.94	1009.60	147.66	148.79	148.79
GRE-3	135.01	947.20	149.05	131.39	131.39
GRE-4	117.13	807.20	111.87	138.16	138.16
SEM	12.69	72.78	16.05	9.87	9.87
*p*-Value	0.3652	0.7389	0.5989	0.6078	0.5039

^a–c^ Mean values bearing different superscript letters within a column are significantly different (*p* < 0.05). CON (unmedicated basal diet), MX (basal diet with bacitracin methylene disalicylate), GRE-1 (basal diet with 0.375% ginger), GRE-2 (basal diet with 0.75% ginger), GRE-3 (basal diet with 1.5% ginger), GRE-4 (basal diet with 3% ginger). DVW (duodenum villi width), DVH (duodenum villi height), DML (duodenum muscle layer), DCD (duodenum crypt depth), DVC (duodenum villi/crypt).

**Table 6 animals-14-01084-t006:** Effect of ginger root extract supplementation on Jejunal villi morphometrics.

	Jejunum Parameters				
Treatments	JVW	JVH	JML	JCD	JVC
Day 6					
CON	70.33	333.77	97.57 ^ab^	77.18	4.39
MX	100.19	435.37	123.61 ^a^	94.79	4.93
GRE-1	109.42	354.16	101.44 ^ab^	97.12	3.96
GRE-2	120.92	443.83	67.48 ^b^	94.79	5.33
GRE-3	81.42	440.94	100.43 ^ab^	89.41	5.05
GRE-4	113.03	413.77	79.94 ^b^	73.47	6.03
SEM	11.38	54.02	8.80	10.26	0.42
*p*-Value	0.1314	0.7557	0.0294	0.7160	0.1369
Day 13					
CON	122.16	517.20	132.46	119.13	4.50 ^b^
MX	126.34	354.90	132.71	95.21	3.89 ^b^
GRE-1	124.28	428.00	106.44	101.10	4.41 ^b^
GRE-2	118.80	490.70	118.59	115.79	4.35 ^b^
GRE-3	113.98	532.70	105.44	94.84	6.16 ^a^
GRE-4	125.70	587.70	137.49	118.47	5.03 ^ab^
SEM	9.42	64.66	10.07	9.38	0.36
*p*-Value	0.3567	0.4011	0.3064	0.4323	0.0297

^a,b^ Mean values bearing different superscript letters within a column are significantly different (*p* < 0.05). CON (unmedicated basal diet), MX (basal diet with bacitracin methylene disalicylate), GRE-1 (basal diet with 0.375% ginger), GRE-2 (basal diet with 0.75% ginger), GRE-3 (basal diet with 1.5% ginger), GRE-4 (basal diet with 3% ginger). JVW (jejunum villi width), JVH (jejunum villi height), JML (jejunum muscle layer), JCD (jejunum crypt depth), JVC (jejunum villi/crypt).

**Table 7 animals-14-01084-t007:** Effect of ginger root extract supplementation on Ileal villi morphometrics.

	Ileum Parameters				
Treatments	IVW	IVH	IML	ICD	IVC
Day 6					
CON	72.39 ^b^	229.19	87.16 ^b^	52.07	4.49
MX	109.00 ^ab^	173.66	63.25 ^b^	45.96	3.81
GRE-1	144.24 ^a^	227.85	85.57 ^b^	56.11	4.29
GRE-2	127.23 ^a^	211.50	118.66 ^a^	51.24	4.19
GRE-3	116.97 ^a^	169.02	79.37 ^b^	53.70	3.46
GRE-4	116.54 ^a^	207.16	89.09 ^b^	54.75	3.92
SEM	11.21	18.35	7.92	4.58	0.23
*p*-Value	0.0370	0.2889	0.0227	0.8387	0.1792
Day 13					
CON	126.07	307.23	152.54	97.10	3.35
MX	152.14	180.60	137.04	79.89	2.73
GRE-1	152.98	235.09	129.45	77.91	3.11
GRE-2	154.92	260.62	118.42	76.52	3.55
GRE-3	122.77	261.27	144.74	80.07	3.44
GRE-4	131.37	265.99	129.60	96.31	2.96
SEM	14.65	29.50	16.98	7.89	0.39
*p*-Value	0.6405	0.2899	0.8802	0.4699	0.8211

^a,b^ Mean values bearing different superscript letters within a column are significantly different (*p* < 0.05). CON (unmedicated basal diet), MX (basal diet with bacitracin methylene disalicylate), GRE-1 (basal diet with 0.375% ginger), GRE-2 (basal diet with 0.75% ginger), GRE-3 (basal diet with 1.5% ginger), GRE-4 (basal diet with 3% ginger). IVW (ileum villi width), IVH (ileum villi height), IML (ileum muscle layer), ICD (ileum crypt depth), IVC (ileum villi/crypt).

**Table 8 animals-14-01084-t008:** Effect of GRE supplementation on goblet cell density of ileum on day 6 and day 13.

	Goblet Cell Density ^1^	
Treatment ^2^	Day 6	Day 13
CON	1.90 ^b^	2.10 ^bc^
MX	2.50 ^ab^	1.60 ^bc^
GRE-1	4.00 ^a^	1.40 ^c^
GRE-2	3.30 ^ab^	1.20 ^c^
GRE-3	2.20 ^b^	2.70 ^ab^
GRE-4	3.50 ^ab^	3.00 ^a^
SEM	0.23	0.14
*p*-Value	0.0353	0.0400

^a–c^ Mean values bearing different superscript letters within a column are significantly different (*p* < 0.05). ^1^ Goblet cell density = number of goblet cells per mm^2^. ^2^ CON (unmedicated basal diet), MX (basal diet with bacitracin methylene disalicylate), GRE-1 (basal diet with 0.375% ginger), GRE-2 (basal diet with 0.75% ginger), GRE-3 (basal diet with 1.5% ginger), GRE-4 (basal diet with 3% ginger).

**Table 9 animals-14-01084-t009:** Linear regression analysis.

Regression	R^2^	*p*-Value
Y^1^ = 1.676 + 0.044 X	0.4670	<0.0010
Y^2^ = 3.3300 − 0.139 X	0.5100	<0.0010
Y^3^ = 7311.73 − 24.33 X	<0.0001	0. 9690
Y^4^ = 1.0110 + 0.211 X	0.0400	0.3860
Y^5^ = 4.570 + 0.295 X	0.1000	0.1730
Y^6^ = 1.5560 + 0.4630 X	0.2530	0.0140
Y^6^ = 6.506 + 0.01 Gb − 0.00004135 BAP + 0.021 JVC − 0.096 MUC2 − 1.725 FCR	0.746	0.0040
Y^7^ = 1.611 + 0.042 Gb + 0.00000655 BAP + 0.002 JVC − 0.025 MUC2	0.035	0.4050
Y^8^ = −3.231 + 0.449 Gb + 0.0000854 BAP + 0.627 MUC2	0.431	0.0350

Y is the response variable, and X is the GRE supplementation, Y^1,2,3,4,5^ = dependent variable (FCR, BWG, BAP, MUC2, JVC, goblet cell density), Y^6^ = BWG, Gb = goblet cell density d13, JVC = jejunum villi/crypt d13, MUC2 = mucin d13, FCR = feed conversion ratio, Y^7,8^ = FCR, Gb = goblet cell density d13, JVC = jejunum villi/crypt d13, MUC2 = mucin d13.

## Data Availability

Data will be made available on reasonable request.

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
