# Peer review of "Effect of Ginger Root Extract on Intestinal Oxidative Status and Mucosal Morphometrics in Broiler Chickens"

_animals, 2024, doi:10.3390/ani14071084_

Round 1
Reviewer 1 Report
Comments and Suggestions for Authors
Manuscript
Effect of Ginger Root Extract on Intestinal Oxidative Status and Mucosal Morphometrics in Broiler Chicken.
Some recommendations are below:
Since all treatments are basal diet not supplemented or supplemented, Please improve treatment description. The basal diet + is repetitive
Improve also in the tables i.e. GRE-0.375%...
Also… in the result tables the treatments present a different abbreviation. This is very confusing, pelase standardize it
This is also divergent between abstract and MM
In the text it’s Not clear that the product was added in the diets, please improve. Same for the abstract
Delete the first and general sentences in the introduction
Paragraph 1 in the introduction should be shortened
L89 delete such as BW, BWG, FI and FCR
L92 – replace & by and
Table 1 and 2 – include all first limiting digestible amino acids
Please provide more information about the tested GRE, composition and details
Table 4 –improve footnotes and numbers. Format (superscript) and they do not match or are not written
Goblet cells per unit area, do you have another unit to present this data? Because the numbers are very low
Please explain better why some analyses were conducted in some ages and were not conducted in other ages d 6, 13, 27 or 41
L441 – please indicate the AGP that was evaluated in this trial
Why you did not perform a regression analysis? Based in your conclusions I would recommend include regression
Comments on the Quality of English LanguageEnglish is fine
Reviewer 2 Report
Comments and Suggestions for Authors
Manuscript number: animals-2921139
Title: Effect of Ginger Root Extract on Intestinal Oxidative Status and Mucosal Morphometrics in Broiler Chicken.
The authors evaluated the effect of ginger root extract (GRE) supplementation on the oxidative status and mucosal development in broiler chickens for 6 weeks, finding that inclusion of ginger root extract up to 1.5% improved growth, reduced oxidative stress, and enhanced mucosal development in broiler chicks. Despite the interesting findings of this work, survival concerns were identified that require the authors' attention. Please see the details below:
Abstract
The number of groups into which laboratory animals are divided should be specified.
Line 23-25: For each treatment, the highest number of laboratory animals should be identified.
Introduction
Line 50: In what ways does ginger influence an increase in immunological response both in vivo and in vitro? Do any mechanisms contribute to this enhancement, and which components of ginger influence this immune response enhancement? Further description is needed.
Line 51: What is the concentration of each substance, and what effect does it have on the experiment?
Line 53-54: Why are only the effects of gingerols on the experiment provided as an illustration? Were additional components of ginger inconsequential to the experiment? Additional clarification is needed.
Materials and Methods
Variability in the experiment can be influenced by incorporating differing quantities of a protein source composition into the diet; therefore, employing a formulation with distinct compositions is preferable to using distinct adjuvant compositions.
Line 98: Why select preserved poultry? Elaborate more on Day 432.
Line 98-100: Specify the number of laboratory animals utilized in each treatment.
Line 240: P = 0.05? Should it be utilized less frequently? Please verify.
Results
Table: For clarification, specify the abbreviations for the parameters beneath the table.
Line 249: To demonstrate effectively with an experiment, it is more appropriate to provide an example of the greatest value rather than the smallest value, to positively demonstrate the efficacy of the experimental results.
Table 5: Adjust the decimal places in the trial table to be equal.
Discussion
Line 350: Complete substance 2. What is the impact of this species? What mechanisms exist to enhance the productivity of animals used in laboratories? Further description is needed.
Comments on the Quality of English LanguageMinor editing of English language required
Reviewer 3 Report
Comments and Suggestions for Authors
The effects of ginger root extract on the growth oxidation state and intestinal mucosa of broilers were studied. However, the readability of the manuscript can be greatly improved to better communicate the importance of your research. After editing and some revisions, I feel that the manuscript is fit for publication. Important weaknesses that need to be satisfactorily addressed are listed below.
1. Line 10 Please replace chicken growth promotion with growth conditions.
2. Line 11 Please unify the title in the whole text, all change to antibiotics.
3. Line 17 Please replace mucosal development with intestinal mucosal development.
4. Line 22 The determination and explanation of pro-oxidant levels were not found in the article, please add.
5. Line 37 Replace the use of antibiotics with the abuse of antibiotics.
6. Line 67 This article is about the study of poultry, please replace the term of birds in the article with poultry.
7. Line 139 Why day 27 and not 28? Please explain.
8. Line 143 Why do you choose these days for sample collection? I don't see any pattern in these days.
9. Line 418 Please elaborate on the cascade reaction.
Reviewer 4 Report
Comments and Suggestions for Authors
Comments to the Authors of manuscript numberanimals-2921139 entitled “Effect of Ginger Root Extract on Intestinal Oxidative Status and Mucosal Morphometrics in Broiler Chicken.”.
This study explores the potential of ginger root extract (GRE) as a substitute for antibiotics in improving chicken production. In a 6-week trial involving broiler chickens, extract was administered at different doses. Results showed improved body weight, body weight gain, and feed conversion ratio in GRE-treated groups compared to control, with the optimal effects observed at 1.5% extract supplementation. Additionally, GRE supplementation improved mucosal development and reduced oxidative stress in the intestines, suggesting its potential as a natural alternative for enhancing chicken health and performance while addressing concerns about antimicrobial resistance in the poultry industry.
1. The introduction provides a comprehensive overview of the poultry industry's challenges, particularly concerning antibiotic use and antimicrobial resistance.
2. Maintenance and nutrition during the first 432 days of life should be presented.
3. L 98- or there is a mistake. 432 one-day-old
4. L 138- for each group, relication?
5. L 143- replication?
6. L 159- what tissue exactly? Duodenum, jejunum- what part of jejunum?
7. L 165- what part of each segment?
8. in the description of material and methods n for each parameters should be given
9. Tbales. N should be given
10. L 257- goblet cell analysis was not described in methods
11. L 297- it should be calculated per the unit of length of villi. Counting goblet cells per unit length of villi, rather than per unit area, is important because it provides a more accurate representation of goblet cell density relative to the surface area available for absorption in the intestine.
12. The description of Figure 1,2,3 should be corrected. N should be given.
13. The description of all figures should be corrected
14. the discussion mentions that ginger root extract contains bioactive components with antioxidant, antimicrobial, and anti-inflammatory properties. Present please specific mechanisms through which these components exert their effects. present more detail on how gingerols, shogaol, and other active compounds in ginger interact with cellular pathways involved in digestion, nutrient absorption, and immune response would enhance understanding.
15. the study mentions the results from previous study regarding the effect of ginger supplementation on bird performance, but discussion on how these findings align or diverge from existing literature would strengthen the argument.
16. Explaining how the changes in villi height, crypt depth, and other morphometric parameters changes correlate with nutrient absorption, gut health, and overall bird performance .
17. the discussion mentions the increase in goblet cell density and mucin production observed with GRE supplementation, it could provide more insight into the functional significance of these changes. Write how enhanced mucin production contributes to intestinal health, pathogen defense, and nutrient absorption would add depth to the discussion. How mucin production was calculated?
Round 2
Reviewer 1 Report
Comments and Suggestions for Authors
Quantitative treatments being evaluated with levels at 0, 0.375, 0.75, 1.5, and 3% and the authors answered that "The statical analysis employed in this work is sufficient". This is not the way we answer a question about regression analysis
Comments on the Quality of English Languageminor editing
Reviewer 2 Report
Comments and Suggestions for Authors
The authors addressed sattify to my comments and further recommendations.
Author Response
Thank you for accepting our responses to your comments.
Reviewer 4 Report
Comments and Suggestions for Authors
Thank you for your answer, which was quite superficial. I did not receive answers to 2 questions. In point 11, it was enough to provide examples of publications where GCs were measured per unit. The reviewer's role is not to answer himself. And the next point 14 was also omitted. If there is previous experience of this type, please discuss it in at least one sentence.
